# A New Technique for Determining Micronutrient Nutritional Quality in Fruits and Vegetables Based on the Entropy Weight Method and Fuzzy Recognition Method

**DOI:** 10.3390/foods11233844

**Published:** 2022-11-28

**Authors:** Xuemei Zhao, Mengdong Zhu, Xiao Ren, Qi An, Junmao Sun, Dazhou Zhu

**Affiliations:** Institute of Food and Nutrition Development, Ministry of Agriculture and Rural Affairs, Beijing 100080, China

**Keywords:** hidden hunger, nutritional quality of fruits and vegetables, method construction, entropy weight method, fuzzy mathematical method

## Abstract

The human body needs nutrients to maintain its regular physiological activity. It requires 40 essential nutrients, including macronutrients (carbohydrates, protein, and fat) and micronutrients (vitamins and minerals). Although macronutrient intake has been improved in China due to people’s increased social awareness, the population’s micronutrient intake remains insufficient. Objective: The current food evaluation system is primarily used to assess macronutrients, while an effective assessment method for micronutrients is still lacking. Fruits and vegetables are low-energy food sources that mainly provide vitamins and minerals and supply the human body with various micronutrients. Methods: In this paper, the entropy and fuzzy recognition methods were used to construct the Vitamin Index (Vitamin Index = Vitamin A Index + Vitamin Comprehensive Index + Vitamin Matching Index) and Mineral Index (Mineral Index = Calcium Index + Mineral Comprehensive Index + Mineral Matching Index) and to evaluate the micronutrient quality of 24 vegetables and 20 fruits. Results: The assessment results showed that Chinese dates displayed the highest Vitamin and Mineral Index among fruits (Vitamin Index = 2.62 and Mineral Index = 2.63), while collard greens had the highest Vitamin Index of the vegetables, at 2.73, and red amaranth had the highest Mineral Index, at 2.74. Conclusions: The study introduces a new method for assessing the nutritional quality of micronutrients, which provides a new idea for assessing the nutrient quality of agricultural products.

## 1. Introduction

### 1.1. Hidden Hunger

The human body needs more than 40 nutrients to maintain normal physiological activities. These include the three macronutrients, namely carbohydrates, proteins, and fat and micronutrients, namely vitamins and minerals. Although the overall nutritional status of the population in China has improved and the remaining problems of hunger have been sufficiently addressed with the development of the economy, there is still a deficiency of micronutrients. The World Health Organization (WHO) and the Food and Agriculture Organization of the United Nations (FAO) define the phenomenon of a lack of micronutrients in the diet as “hidden hunger”. Hidden hunger refers to a nutritionally unbalanced diet or a deficiency of certain vitamins or minerals that results in hunger symptoms of hidden nutritional need due to an insufficient intake of micronutrients [1].

Hidden hunger can lead to congenital disabilities, stunted growth, increased mortality in children and pregnant women, a weakened immune system, and various other symptoms that severely affect human health. Studies have shown that approximately 821 million people worldwide are chronically overtly hungry [2]. However, more than two billion people are affected by hidden hunger [3], which is currently increasing. In 2015, the WHO pointed out that nearly five billion people in the world suffer from various micronutrient deficiencies. Of these, about two billion suffer from an iron and iodine deficiency. Zinc deficiency is found in 17.3% of the total global population. In addition, about 230 million preschool children suffer from vitamin A deficiency. The International Food Policy Research Institute (IFPRI) points out, in the Global Nutrition Report 2016, that currently about one-third of the global population is undernourished, leading to growth retardation, individual weight loss, obesity and deficiencies of essential vitamins and minerals. In 2000, the WHO pointed out that vitamin A, iodine, iron, and zinc deficiencies are a serious health risk [4]. According to the International Zinc Association, approximately 49% of the world’s population is affected by zinc deficiency, which manifests itself through symptoms such as slow growth, late sexual maturity, hypogonadism, hair loss, slow wound healing, and immune deficiency, and impairs normal physiological activities in the human body [5]. Micronutrients intake is also inadequate in China. Almost 300 million inhabitants are currently suffering from hidden hunger [6].

Although the demand for micronutrients is very low, the importance of micronutrients for health was recognized late, leading to some neglect of micronutrients. However, micronutrients are crucial for maintaining human health, and a long-term deficiency of them can lead to various health problems [7]. Shouhang Chen [8] found that micronutrient deficiency in the perinatal period can lead to anemia in pregnant women, premature delivery, weak babies and fetal development delays. Adequate supply of micronutrients such as iron, zinc, vitamin A, and vitamin C during the perinatal period is closely related to pregnancy outcomes and has a greater impact on the health of pregnant women and infants. Jin Yang et al. [9] conducted a meta-analysis of vitamin D and sepsis. The results showed that the risk of sepsis was 1.68 times higher in patients with vitamin D deficiency than in patients with normal vitamin D levels (95% CI: 1.58 to 1.80).

### 1.2. Vitamin Function and Intake Status

Vitamins are low-molecular organic compounds necessary for the maintenance human life activities. They are neither the main material of which the various tissues are composed, nor are they the source of physiological energy in the body. Vitamins are essential for human metabolism and energy supply. For example, vitamin A can be a light-sensitive substance in photoreceptor cells, while vitamin C has an antioxidant effect, caused by its ability to scavenge free radicals and improve the absorption of folic acid. Vitamins generally occur in their natural form or as precursors that can be used by the body. Although a small amount is required, most vitamins can neither be synthesized in the body nor stored in large amounts in the body’s tissues and must be taken in through the diet. Although niacin and vitamin D can be synthesized by the body autonomously, and intestinal bacteria can produce vitamin K and biotin, the subsequent quantity does not completely cover the requirements.

According to data published in the 2013 Report on the Nutrition and Health Status of Chinese Residents, the current average daily intake of energy, protein, fat, and carbohydrates per person in urban and rural areas are 2162.3 kcal (9047.1 kJ), 64.25 g, 79.7 g, and 299.2 g, respectively. Compared with the recommended nutrient intake, it has met or exceeded the Recommended Nutrient Intake (RNI). However, the average daily vitamin intake per person in urban and rural areas is far below the RNI or Adequate Intake (AI), as shown in Table 1 and Appendix A.

Table 1 and Appendix A show the severity of vitamin deficiency among Chinese residents. The daily per capita intake of vitamin A in urban and rural areas is only 291.5 μg/Retinol Activity Equivalents (RAE), while the daily vitamin A intake of school-age children and the elderly is 351.9 μg/RAE and 398.8 μg/RAE, respectively, less than half the RNI. In China, vitamin A intake is below the RNI for 95.2% of people and above 85% for school-age children and the elderly. Although the proportion of people with low intakes of vitamins B1, B2, and C is also high, their intakes reach more than half RNI. Therefore, vitamin A is considered the greatest cause of hidden hunger.

### 1.3. Mineral Function and Intake

Human tissue contains approximately 92 natural components. The types and content of these elements in different people are linked to their food consumption and residential environment. In addition to the elements of carbon, hydrogen, and nitrogen components that make up organic compounds, others are called minerals, also called inorganic salts or ash. Minerals can be divided into macro and trace elements according to their content levels in the body. Mineral levels higher than 0.01% of a person’s body weight are known as major or macro elements and include calcium, phosphorus, sodium, potassium, sulfur, chlorine, and magnesium, while minerals lower than this level represent microelements or trace elements. Minerals cannot be synthesized in the human body and need to be acquired from external sources. Some minerals are excreted from the body through metabolic processes, including urine, sweat, hair, nails, and epithelial cells. Therefore, constant food supplementation is necessary in order to obtain the necessary minerals involved in many physiological and metabolic processes in the human body. For example, calcium is the major element in bones while transmitting cellular information and regulating enzymatic activity. Zinc is known as a life element, participating in protein synthesis, cell growth, division, and differentiation. A lack of zinc may cause RNA, DNA, and protein synthesis challenges, reduce cell division, and lead to growth arrest.

Compared to protein, carbohydrate, and fat intake, mineral intake is considerably lower than the RNI in China and does not meet the requirements for human physiological activities. The daily mineral intake for urban and rural Chinese residents is shown in Table 2 and Appendix A.

Table 2 and Appendix A show that, with the exception of iron, phosphorus, and zinc (male) intake, the consumption of other minerals is lower than the RNI. The calcium intake is only 364.3 mg, which is less than half the RNI. Furthermore, 98.5% of Chinese residents display insufficient calcium intake. This value in school-aged children and the elderly is close to the national average rate, indicating that nearly 98 out of 100 people in China are calcium deficient, necessitating an urgent resolution. Current data show that hidden hunger affects 300 million in China [10], making it a significant challenge.

### 1.4. The Nutritional Value of Fruits and Vegetables

Many vegetables are typically rich in vitamins, minerals, and dietary fiber and low in protein and fat. Vegetables display good sensory properties due to the presence of organic acids, aromatic substances, and pigments, promoting appetite, digestion, and diversification. In addition, vegetables also contain various phytochemicals responsible for several biological effects that are beneficial to human health. Increasing vegetable intake can maintain health and effectively reduce the risk of chronic conditions, such as cardiovascular disease, cancer, and diabetes.

Wenzhe Hou [11] used high-performance liquid chromatography-mass spectrometry and gas chromatography-mass spectrometry to detect 16 phenolic acids and 11 sterols in ten types of vegetables. Mengya Deng [12] used the Nutritional Quality Index (INQ) to analyze the calcium, potassium, sodium, magnesium, iron, zinc, copper, manganese, selenium, and iodine levels in ten dark-colored vegetables and ten light-colored vegetables collected from the Shenzhen farmer’s market. The results indicated that the potassium and magnesium INQ in all of the vegetables was >1, while the nutritional mineral value of the dark-colored vegetables was higher than the light-colored vegetables. Therefore, it is better to increase the proportional intake of dark-colored vegetables. Gang Peng [13] examined the content of nine nutrients in vegetables and the DPPH and ABTS activity in vegetable juice.

The carbohydrate level is between approximately 5% and 30% higher in fruits than in vegetables and is mainly present in monosaccharides or disaccharides. Fruits are rich in organic acids, stimulating digestive gland secretion and facilitating food digestion. Furthermore, there are various vitamins in fruits, which are the coenzymes of most enzymes. For example, vitamin B6 can participate in amino acid decarboxylation, tryptophan synthesis, sulfur-containing amino acid metabolism, and unsaturated fatty acid metabolism. In addition, fruits contain phytochemicals, such as flavonoids, aromatic substances, coumarin, and D-lemon terpene that participate in biological activities and are beneficial to human health.

Simona et al. [14] studied the main nutritional characteristics of 41 cherry varieties from central Italy, indicating a sorbitol content as high as 44.2 mg/g, while the malic acid level was 48.4 mg/g, confirming the high nutritional value of cherries. Feng Wang [15] found that polyphenols, dietary fiber, and other phytochemicals in fruits and vegetables can interact with intestinal bacteria, changing the structure of intestinal flora and positively impacting human physiological functions. Moreover, intestinal flora can help the body absorb the functional ingredients and nutrients in fruits and vegetables. Yingqian Wang et al. [16] analyzed the vitamin C, polyphenol, dietary fiber, ash, total sugar, reducing sugar, protein, and amino acid content in nine pineapples varieties, showing significant variation in the nutritional value of the different strains. Yi He [17] analyzed the total sugar, total acid, fiber, protein, fat, ash, vitamins, and amino acids in sea-buckthorn juice from Inner Mongolia, revealing that it contained γ-aminobutyric acid (CABA), accounting for 11.6% of the total free amino acids.

### 1.5. Methods for Evaluating the Nutritional Quality of Food

Since the reform and opening of the Chinese economy, the perspective has shifted from addressing the challenge of supplying food and clothing to enhancing nutrition and balancing nutrient intake. Therefore, based on the concept of a reasonable diet, balanced nutrition has attracted significant attention in nutritional research [18].

Existing food evaluation methods include the INQ [19], Nutrient Profiling (NP) [20], Nutrient Balance Concept (NBC) [21], Index of Nutrient Balance (INB) [18], Nutrient Reference Values (NRV) [22], Amino Acid Score (AAS), Chemical Score (CS), Digestibility Corrected Amino Acid Score (PDCAAS), and essential fatty acid content ratio.

Bao Luo et al. [23] evaluated the amino acid and fatty acid composition and the composition ratio of Japanese flounder using the standard AAS model and egg protein model. Xilin Xu et al. [24] conducted a nutritional evaluation of egg-tofu and milk-tofu and compared the amino acid composition and composition ratio between these products. The results indicated that adding eggs to tofu can increase the protein content and nutritional value, elevating the level of sulfur-containing amino acids. Smruthi [25] evaluated the nutritional quality of the total sugars, organic acids, and phenols in five nectarine varieties from the northwestern and northeastern Himalayas. Weimin Zhang et al. [26] analyzed the protein and functional components in noni fruits and leaves while conducting a nutritional evaluation via AAS.

These methods are not suitable for evaluating the nutritional quality of fruits and vegetables and are more often used to assess the relationship between macronutrients and energy. Although micronutrients do not directly provide energy, they are essential for life-sustaining activities. Evaluating vitamins and minerals in fruits and vegetables using the methods mentioned above presents certain limitations. To resolve the challenge of unclear concepts, unclear references, and inconsistent indicators presented by the current evaluation methods of the nutritional value of fruits and vegetables, Zhiqin Zhou [27] proposed the “three degrees method” to assess fruits, which involves the degree of diversity (DD), degree of match (DM), and degree of balance (DB). This technique evaluates the overall nutritional value in a comprehensive, systematic, and standardized way. It is a preliminary attempt to assess the overall nutritional value of fruits.

The current daily energy intake in China meets or exceeds the RNI. Therefore, attempts should focus on addressing hidden hunger and high-calorie dietary challenges. Fruits and vegetables are low in calories and rich in dietary fiber, providing a variety of vitamins, minerals, and phytochemicals. Reducing energy intake and increasing the consumption of fruits and vegetables can effectively control the problem of excess energy while improving the current micronutrient deficiency in China. However, most existing research involving fruits and vegetables focuses on analyzing the types and content of micronutrients and phytochemicals. The nutritional value of vitamins and minerals in fresh produce has not been analyzed in-depth, making it challenging to provide accurate intake recommendations. As some fruits and vegetables have lower calorie content than staple foods, such as bananas, which have higher calorie content than leafy vegetables but are still lower than rice at 89 kcal/100 g and 347 kcal/100 g, respectively, according to the 2016 Chinese Food Composition Table, they do not provide much energy compared to staple foods. Therefore, it is necessary to reasonably consider the nutrients that fruits and vegetables provide to the human body, rather than analyzing all nutrients. Based on the three degrees method, this paper considers the actual vitamin and mineral content in fruits and vegetables to determine the ratio required to sufficiently meet the nutritional requirements of humans.

## 2. Materials and Methods

### 2.1. Data

The vitamin and mineral data of the fruits and vegetables used in this article are derived from the 2016 Chinese Food Composition Table. According to the 2016 Chinese Food Composition Table classification for fruits and vegetables, six fruits and five vegetables were selected. The fruits included apples, pears, peaches, Chinese dates, apricots, cherries, grapes, pomegranates, blackcurrants, kiwis, strawberries, oranges, kumquats, lemons, pineapples, bananas, lychees, mangoes, watermelons, and cantaloupes. The vegetables included white radishes, carrots, Chinese celery, peas, lentils, kidney beans, eggplants, tomatoes, peppers, cucumbers, pumpkins, Chinese cabbage, wuta-tasi, rapes, collard greens, broccoli, spinach, celery, lettuce, red amaranth, mustard greens, yam, taro, and potatoes. The vitamin and mineral content data are shown in Appendix A.

### 2.2. Vitamins and Minerals Index

Analysis indicated that the vitamin and mineral intake values were both lower than the RNI, particularly vitamin A and calcium. Therefore, three aspects are considered in real conditions when establishing the model. firstly, the vitamin A and calcium intake in China is considerably lower than the RNI. Therefore, the Vitamin A and Calcium Indexes are established in the evaluation model. Secondly, considering the actual content of vitamins and minerals in fruits and vegetables, the Vitamin Comprehensive Index and Mineral Comprehensive Index are established. Last but not least, it is also necessary to link the actual vitamin and mineral content in fruits and vegetables with human deficiencies, proposing the matching index concept and establishing the Vitamin Matching Index and Mineral Matching Index.

These three indexes represent the vitamin A (calcium) content, the comprehensive nutritional quality of vitamins (minerals) in fruits and vegetables, and the matching degree of vitamins (minerals) in fruits and vegetables with deficiencies in Chinese people. The formula is as follows.

### 2.3. Vitamin A Index and Calcium Index

Vitamin A Index and Calcium Index calculation. The vitamin A or calcium content in the 24 vegetables and 20 fruits are sorted from smallest to largest, after which the first quartile, median, and third quartile are calculated. A Vitamin A Index and Calcium Index less than or equal to the first quartile is assigned 0.25. If they are less than or equal to the median, they are assigned 0.5. If they are less than or equal to the third quartile, they are assigned 0.75, while the remainder is assigned a value of 1.

### 2.4. Vitamin Comprehensive Index and Mineral Comprehensive Index

Vitamin Comprehensive Index and Mineral Comprehensive Index calculation. The entropy weight method is used to calculate the nutritional quality of the vitamins and minerals in the vegetables and fruits. Entropy involves thermodynamics. Using probabilistic statistical methods, entropy can be used to measure insufficient system information or chaotic disorder [28]. Lower systematic information indicates more disorder, while a higher level of systematic information demonstrates more order. The view that “entropy will be the first law of the entire science” (Einstein) has been verified [29]. The theory of information entropy is currently widely used in various fields such as tourism, agriculture, safety production, and medical and health, indicating its significant value and applicability.

The entropy method is suitable for evaluating the “good and bad” of a set of objects according to multiple indicators [30]. Xu Zhang [31] used the entropy method to comprehensively analyze the various indicators of different proportions of potato noodles. Zhiheng Zhou et al. [32] evaluated the satisfaction level based on the entropy weight method and built a comprehensive fuzzy evaluation model of the satisfaction level. Yunqian Ma et al. [33] studied the nutritional value of potatoes based on the entropy-weighted rank-sum ratio and comprehensively evaluated the nutritional value of seven types of potatoes in Gansu.

The entropy weight method is used to evaluate the comprehensive nutritional quality of vitamins and minerals in fruits and vegetables. The calculation process is as follows.

Data standardization. With n samples and k indicators, *X_ij_* represents the measured value of the *j* indicator and the *i* sample (*i* = 1, 2, …, n; *j* = 1, 2, …, k), the *y_ij_* is the standardized value of each indicator, max (*X_j_*) is the maximum value of the indicators, and min (*X_j_*) is the minimum value of the indicators.
$$y_{ij}=\frac{X_{ij}-min(X_{j})}{max(X_{j})-min(X_{j})}$$

Information entropy. *E_j_* is the information entropy of each indicator value, and if *p_ij_* = 0, the $\underset{y_{ij}}{lim}=p_{ij}lnp_{ij}=0.$
$$E_{j}=-ln{\left(n\right)}^{-1}\sum_{i=1}^{n}p_{ij}lnp_{ij}\;p_{ij}=\frac{y_{ij}}{\sum_{i=1}^{n}y_{ij}}$$

Calculating the weight of each indicator.
$$W_{j}=\frac{1-E_{j}}{k-\sum E_{j}}$$

Calculating the comprehensive index.
$$Z_{i}=\overset{k}{\underset{j=1}{\sum }}X_{ij}W_{j}$$

### 2.5. Vitamin Matching Index and Mineral Matching Index

Calculation of the Vitamin Matching Index and Mineral Matching Index. Fuzzy mathematics is used to calculate the matching degree between the vitamin and mineral content in fruits and vegetables and the deficiency levels. Fuzzy mathematics is a mathematical discipline effective for studying uncertain and fuzzy phenomena. The basic idea of fuzzy mathematics is to use accurate mathematical methods to describe and model fuzzy concepts in real life to provide clarification. Fuzzy mathematics theory includes many applications, including fuzzy pattern recognition, fuzzy cluster analysis, fuzzy comprehensive evaluation, fuzzy optimization, fuzzy decision-making, and fuzzy prediction.

The amino acid fuzzy recognition method uses Lance and Williams distance to calculate the closeness between the essential amino acid content of the evaluation object and the standard amino acid pattern value. The closeness can reflect the evaluated object to the model spectrum. A value close to 1 indicates a higher DM [34].

The fuzzy recognition method can be used to calculate the matching degree of the micronutrients in fruits and vegetables and the level of deficiency in humans. The calculation formula is as follows:$$U\left(a,u_{j}\right)=1-C\sum_{i=1}^{n}\frac{\left|a_{k}-u_{ij}\right|}{a_{k}+u_{ij}}$$

*U* represents the DM.

*C* is a constant, 0.09.

*a_k_* represents the standard value. This subject represents the micronutrient deficiency value or the recommended intake value. 

*u_ij_* represents the value of the corresponding micronutrient in each kind of fruit and vegetable, *i* = type of micronutrient, *j* = type of evaluation fruits and vegetables.

This model is based on the principle of the amino acid fuzzy recognition method and uses Lance and Williams distance to characterize the matching relationship between the actual vitamin and mineral content in the fruits and vegetables and the human deficiency level, establishing the Vitamin Matching Index and the Mineral Matching Index.

### 2.6. Nutrient-Rich Food Index (NRF9.3)

NP refers to a method of grading and evaluating foods based on nutrient content [20]. Using the NP model to evaluate different foods can guide consumers into making informed decisions about healthy foods. The NP model is currently used to improve public health awareness and reduce non-communicable diseases (NCDs), and it has gradually attracted widespread attention [35], including in the United States, Australia, New Zealand, Canada, Germany, Belgium, the Netherlands, Sweden, and France, which have established more than 20 NP models and applied them to health recommendations, consumption guidance, dietary guidelines, nutrition labels, and nutrition and health claims [36].

NRF9.3 has been widely used for evaluating the nutritional quality of food. The model can evaluate foods according to the different nutrients and rank the final scores [37,38]. We selected NRF9.3 to evaluate 24 vegetables and 20 fruits. The recommended nutrients include protein, dietary fiber, vitamin A, vitamin C, vitamin E, calcium, iron, magnesium, and potassium, while the restrictive nutrients include sodium, saturated fat, and total sugars [39,40].
$$\begin{array}{l} NRF9.3={NR}_{9}-{ILM}_{3}\\ {NR}_{9}=\sum_{i=1}^{9}\left(Amount\;i/DVi\times 100\right)\\ {ILM}_{3}=\sum_{i=1}^{3}\left(Amount\;i/MRVi\times 100\right) \end{array}$$

*DVi* represents the recommended daily amount of this nutrient.

*MRVi* indicates the maximum recommended daily amount of this nutrient.

*Amount i* represents the actual content of the nutrient in a 100 g sample.

## 3. Results

### 3.1. The Vitamin Index and Mineral Index Results in the Fruits

The Vitamin Index (Vitamin Index = Vitamin A Index + Comprehensive Index + Matching Index) and Mineral Index (Mineral Index = Calcium Index + Comprehensive Index + Matching Index) of 20 types of fruit were calculated, as shown in Table 3; more details are provided in Appendix A.

The top five fruits in the Vitamin Index were Chinese dates, blackcurrants, mangoes, cantaloupes, and kumquats, while the top five fruits in the Mineral Index were Chinese dates, lemons, blackcurrants, kumquats, and cherries. Chinese dates ranked first in both the Vitamin and Mineral Indexes, with values of 2.62 and 2.63, respectively. Blackcurrants ranked second and third in the Vitamin Index and Mineral Index, with scores of 2.08 and 2.58, respectively.

Chinese dates, apricots, kumquats, cantaloupes, and mangoes all scored 1 point in the Vitamin A Index, while Chinese dates ranked first in the Comprehensive Index with 1 point. The score interval did not differ much in the Matching Index. Lemons were the highest at 0.68, exceeding mangoes (0.67) and cherries (0.67). Chinese dates ranked first in the Vitamin A and Comprehensive Indexes and sixth in the Matching Index. Lemons scored higher in the Matching Index, indicating that they are more suitable for addressing the current vitamin deficiency in China. However, lemons did not score high in the Vitamin Index. Although the Matching Index indicated that lemons were highly compatible with nutritional requirements, the vitamin A content in lemons was only 3 μgRAE/100 g, which was substantially lower than Chinese dates, apricots, kumquats, cantaloupes, and mangoes. The DM represented the relationship between the vitamin content and the human deficiency. The vitamin content of some fruit may be much higher or lower than the human deficiency level, while some deviations may be evident in the calculation results. Therefore, it is necessary to analyze the vitamins according to the results.

The Mineral Index rankings differed slightly from the Vitamin Index score results, with the top five fruits being Chinese dates, lemons, blackcurrants, kumquats, and cherries. Blackcurrants rank second in the Vitamin Index and third in the Mineral Index. Chinese dates, blackcurrants, kumquats, and lemons all rank first in the Calcium Index, while only blackcurrants scored 1 point in the Comprehensive Index, followed by Chinese dates and lemons, both receiving 0.98. Chinese dates scored the highest in the Matching Index at 0.65. In addition, the kumquats ranked in the top five in both the Vitamin and Mineral Indexes. The final ranking results showed that, compared with other fruits, Chinese dates, blackcurrants, and kumquats ranked high in both the Vitamin and Mineral Indexes, indicating that these three fruits were superior in meeting the nutritional needs of the human body. Zidan Zhao [40] examined the nutritional components and functional properties of Chinese dates, revealing that they contained cyclic adenosine phosphates, triterpenoids, flavonoids, polysaccharides, and vitamin C. Jianxin Fu [41] investigated the health benefits of Chinese dates, analyzing nutrients, such as polysaccharides, cyclic nucleotides, dietary fiber, organic acids, and vitamins. Xizhen Hu [42] reviewed the nutritional value of blackcurrants, revealing that their vitamin C content was higher than most fruits, while they were rich in mineral elements, such as potassium, calcium, phosphorus, and magnesium. Hunt et al. [43] examined blackcurrant extract in a random double-blind experiment and found that these berries are rich in anthocyanins, which protect against exercise-induced muscle damage (EIMD) and promote the rapid recovery of muscle function. Therefore, based on the comprehensive evaluation results, the Vitamin and Mineral Index scores of Chinese dates and blackcurrants were among the highest. Compared with other fruits, they can provide more vitamins and minerals to meet the nutritional needs of the human body.

### 3.2. The Vitamin Index and Mineral Index Results in the Vegetables

The Vitamin Index (Vitamin Index = Vitamin A Index + Comprehensive Index + Matching Index) and Mineral Index (Mineral Index = Calcium Index + Comprehensive Index + Matching Index) of 24 types of vegetables were calculated, as shown in Table 4; further details are provided in Appendix A.

Collard greens (2.74), carrots (2.59), spinach (2.41), mustard greens (2.13), and red amaranth (2.08) ranked as the top five vegetables in the Vitamin Index, while red amaranth (2.75) Chinese celery (2.53), (2.51), collard greens (2.47), and spinach (2.46) were the top five vegetables in the Mineral Index. Therefore, collard greens, spinach, and red amaranth rank high in both indexes.

Carrots, rape, collard greens, spinach, red amaranth, and mustard greens are rich in vitamin A, all scoring 1 point in the Vitamin A Index, while collard greens scored highest in the Comprehensive Index, with 1 point. Spinach scored highest in the Matching Index at 0.75, followed by collard greens at 0.74. Chinese celery, Wuta-Tasi, rape, collard greens, spinach, and red amaranth scored 1 point in the Calcium Index. Only red amaranth scored 1 point in the Comprehensive Index, followed by Wuta-Tasi (0.80) and Chinese celery (0.79), while spinach scored highest in the Matching Index at 0.79. The scores indicated that some vegetables were higher in certain characteristics than others. For example, spinach ranks at the top of the Vitamin and the Mineral Matching Indexes, but not the Vitamin and Mineral Indexes. This could be attributed to the fact that the model considered human deficiency as well as the real vitamin and mineral content in the fruits and vegetables. The Matching Index can only reflect the degree to which the content meets human needs, while other facets should also be considered. Chao Wang et al. [44] analyzed various vegetables and revealed that collard greens were rich in mineral elements, including potassium, iron, zinc, strontium, copper, magnesium, fiber, and β-carotene. Xiaoqi Tao et al. [45] studied the nutritional value and processing characteristics of collard greens, finding an abundance of vitamin C and minerals, particularly calcium, iron, and phosphorus. Mengya Deng et al. [12] used the INQ method to compare the mineral content of dark- and light-colored vegetables, revealing that the nutritional quality of red amaranth was the highest.

### 3.3. Comparison with NRF9.3 Model

The NRF9.3 model was used to evaluate the fruits and vegetables, and the scores are shown in Table 5.

As shown in Table 5, Chinese dates, blackcurrants, kiwis, strawberries, and kumquats ranked as the top five of all the fruits, while peppers, collard greens, spinach, red amaranth, and mustard greens were the top five vegetables. A comparison showed that Chinese dates scored the highest in the Vitamin and Mineral Indexes, as well as the NRF9.3 model, indicating the superiority of its nutritional quality. Furthermore, it also demonstrated that the model in this paper was similar to the NRF9.3 model, validating the Vitamin and Mineral Indexes. Moreover, collard greens, spinach, and red amaranth all ranked in the top five in the NRF9.3 model, as in the Vitamin Index and Mineral Index, demonstrating the accuracy of this model. However, slight differences were evident. For example, peppers appear in the middle in the Vitamin and Mineral Indexes but at the top in the NRF9.3 model due to its higher vitamin C and fiber content and lower sodium content. Analysis of the Vitamin Index score indicated that peppers ranked seventh, and their score was higher than most vegetables. Therefore, the nutritional vitamin value of peppers was higher. However, the Mineral Index score of the peppers was not remarkable, especially regarding the calcium content, which is about 9 mg/100 g.

## 4. Discussion

Fruits and vegetables are low in fat but rich in fiber and micronutrients, including many vitamins and minerals [46]. A new method (Vitamin Index and Mineral Index) was established to comprehensively evaluate the vitamins and minerals in fruits and vegetables according to their characteristics, combined with Chinese intake statistics. Each index involves three dimensions: (1) the most deficient vitamins and minerals; (2) the comprehensive quality of the vitamins or minerals in fruits and vegetables; (3) the matching degree between the content and deficiency. These three dimensions correspond to three indicators: Vitamin A Index/Calcium Index, Vitamin/Mineral Comprehensive Index, and Vitamin/Mineral Matching index. The results indicated that Chinese dates presented the highest Vitamin and Mineral Index scores, while collard greens scored the highest in the Vitamin Index and red amaranth ranked highest in the Mineral Index. The average Vitamin and Mineral Index scores were 1.43 and 1.61 for the fruits and 1.49 and 1.71 for the vegetables, which was verified by the NRF9.3 model results, with average scores of 64.74 and 72.32, respectively. Consequently, vegetables provide more vitamins and minerals than fruit. However, fruits are rich in phytochemicals, such as carotenoids, phenolic compounds, plant sterols, protease inhibitors, terpenes, sulfides, and phytic acid [47]. Therefore, it cannot be stated that the nutritional value of fruits is lower than vegetables. Although the vitamins and minerals provided by vegetables exceed that of fruits, analysis of overall diet structure indicates that fruits and vegetables are both vital for human health.

The NRF9.3 scoring results were similar to those of the Vitamin Index and Mineral Index. Chinese dates, blackcurrants, and kumquats ranked in the top five of Vitamin Index and Mineral Index, while collard greens, red amaranth, and spinach ranked in the top five as well, confirming that their nutritional vitamin and mineral quality exceeded most other fruits and vegetables, reflecting the similarity with the NRF9.3 model. Compared with the NRF9.3 model, the Vitamin and Mineral Indexes consider the nutritional quality of the vitamins and minerals in fruits and vegetables, as well as the actual intake in China. However, the NRF9.3 model takes protein, dietary fiber, energy, and sugar into account, which is absent from the model built in this article, and micronutrients are the main nutrients provided by the fruits and vegetables. Although this model does not consider protein, dietary fiber, and energy, it captures the main characteristics of fruit and vegetable nutrition. Compared with processed foods, which require a comprehensive examination of nutrients, such as protein, carbohydrates, energy, minerals, and vitamins, the main purpose of fruits and vegetables is to provide vitamins and minerals rather than energy. For some foods, such as potatoes, sweet potatoes, and other foods with slightly higher starch content, there may be deviations when using the NRF9.3 model, resulting in lower scores. Although it is generally believed that they have higher calories than leafy vegetables, their calorie content is exceedingly low compared to staple foods, such as rice or wheat.

In addition to fruits and vegetables, micronutrients can also be obtained from some staple foods (rice, potatoes, and sweet potatoes) or meat (beef, pork, lamb, and fish). However, these foods mainly provide carbohydrates and proteins, which are not good sources of micronutrients. Many consumers prefer to consume staple foods, such as refined, processed rice and noodles. Compared with coarse grains, such as brown rice and germ rice, the total nutrient loss rate is approximately 66% during the manufacturing process [48]. Therefore, the Vitamin and Mineral Index models specifically evaluate the micronutrients in fruits and vegetables, demonstrating the nutritional quality of micronutrients in different fruits and vegetables while promoting the benefits of fresh produce to customers.

## 5. Conclusions

Food provides energy and a variety of nutrients to the human body to address a variety of physiological needs. Although the problem of insufficient energy intake in China has been improved, the consumption of micronutrients is far from sufficient, with the overall intake of vitamins and minerals being lower than the RNI. Vitamin A and calcium deficiency is the most serious challenge, with intake being less than half of the recommended intake and with a deficiency rate close to 100%. The current food evaluation system focuses primarily on evaluating macronutrients and lacks an effective method for assessing micronutrients. The analysis of the current data analysis of vitamin and mineral intake data indicates that vitamin A and calcium are considered important indicators of assessment. The micronutrient Comprehensive Index is calculated using the entropy weight method, while the Matching Index of the volume of micronutrients in food and human deficiency is evaluated and calculated using the fuzzy recognition method. Fruits and vegetables are low in calories and can supply a variety of vitamins and minerals to the human body, which are ideal food sources of micronutrients.

However, this model presents certain limitations. For example, the Vitamin and Mineral Indexes only consider vitamin A and calcium. Therefore, the weight of the components in certain fruit and vegetables with outstanding nutritional content must be considered separately. For example, for some people who are severely deficient in vitamin C, prickly pear is the best choice compared to other types of fruits. However, this model uses the average demand as the threshold and does not consider the particular needs of individuals. Therefore, future studies should consider the variation in the needs of the population to further improve the evaluation model.

## Figures and Tables

**Table 1 foods-11-03844-t001:** The average daily intake of vitamins per standard person for urban and rural residents across the country from 2010 to 2013. The intake of vitamin E uses AI. The data source population represents the reference intake of dietary nutrients for people aged 18–50.

	Vitamin A μg/RAE/d		Vitamin B1 mg/d		Vitamin B2 mg/d		Niacin mgNE/d		Vitamin C mg/d	Vitamin E mgα-TE/d
Average	291.5		0.9		0.8		14.3		80.1	8.5^③^
City	334.9		0.9		0.8		14.9		84.9	9.5
Rural	249.8		1.0		0.7		13.6		75.4	7.6
Reference Intake	male	female	male	female	male	female	male	female	100	14
	800	700	1.4	1.2	1.4	1.2	15	12		

**Table 2 foods-11-03844-t002:** The mineral intake of urban and rural residents in China from 2010 to 2013 (mg/d). The selenium intake is expressed in μg, while the recommended potassium intake value is expressed as AI.

	Calcium	Magnesium	Potassium	Phosphorus	Iron		Zinc		Selenium
Average	364.3	283.4	1610.4	950.6	21.4		10.7		44.4
City	410.3	279.6	1654.3	964.3	21.8		10.6		46.9
Rural	320.1	290.6	1567.9	937.1	21.1		10.7		42.1
RNI	800	330	2000	720	male	female	male	female	60
					12	20	12.5	7.5	

**Table 3 foods-11-03844-t003:** The Vitamin Index and Mineral Index results in the fruits.

	Sorting			
	Vitamin Index		Mineral Index	
Apples	2.62	Chinese Dates	2.63	Chinese Dates
Pears	2.08	Blackcurrants	2.61	Lemons
Peaches	1.99	Mangoes	2.58	Blackcurrants
Chinese Dates	1.94	Cantaloupes	2.15	Kumquats
Apricots	1.88	Kumquats	1.87	Cherries
Cherries	1.76	Apricots	1.80	Apricots
Grapes	1.67	Kiwis	1.79	Bananas
Pomegranates	1.55	Oranges	1.71	Oranges
Blackcurrants	1.52	Cherries	1.65	Strawberries
Kiwis	1.40	Watermelons	1.58	Pineapples
Strawberries	1.22	Pomegranates	1.40	Pomegranates
Oranges	1.18	Bananas	1.39	Mangoes
Kumquats	1.08	Apples	1.24	Watermelons
Lemons	1.05	Strawberries	1.24	Cantaloupes
Pineapples	1.05	Lychees	1.23	Grapes
Bananas	1.03	Lemons	1.20	Pears
Lychees	0.93	Pineapples	1.13	Lychees
Mangoes	0.90	Peaches	1.09	Kiwis
Watermelons	0.86	Grapes	1.05	Peaches
Cantaloupes	0.85	Pears	0.90	Apples

**Table 4 foods-11-03844-t004:** The Vitamin Index and Mineral Index Results of the vegetables.

	Sorting			
	Vitamin Index		Mineral Index	
White Radish	2.74	Collard Greens	2.75	Red Amaranth
Carrots	2.59	Carrots	2.53	Chinese Celery
Chinese Celery	2.41	Spinach	2.51	Wuta-Tasi
Peas	2.13	Mustard Greens	2.47	Collard Greens
Lentils	2.08	Red Amaranth	2.46	Spinach
Kidney Beans	2.06	Rape	2.40	Rape
Eggplant	1.91	Peppers	1.81	Broccoli
Tomatoes	1.72	Wuta-Tasi	1.80	Kidney Beans
Peppers	1.58	Pumpkins	1.80	Peas
Cucumbers	1.54	Peas	1.80	Lentils
Pumpkins	1.54	Broccoli	1.72	Chinese Cabbage
Chinese Cabbage	1.49	Tomatoes	1.69	White Radish
Wuta-Tasi	1.23	Chinese Cabbage	1.54	Mustard Greens
Rape	1.15	Eggplant	1.46	Carrots
Collard Greens	1.14	Kidney Beans	1.45	Celery
Broccoli	1.12	Cucumbers	1.44	Tomatoes
Spinach	1.10	Yam	1.36	Eggplant
Celery	1.08	Lentils	1.36	Potatoes
Lettuce	0.93	Potatoes	1.30	Cucumbers
Red Amaranth	0.87	White Radish	1.24	Peppers
Mustard Greens	0.86	Lettuce	1.23	Yam
Yam	0.83	Chinese Celery	1.04	Pumpkins
Taro	0.81	Celery	0.95	Lettuce
Potatoes	0.80	Taro	0.92	Taro

**Table 5 foods-11-03844-t005:** The NRF9.3 score results of 20 fruits and 24 vegetables.

Fruits	Score	Vegetables	Score
Chinese Dates	272.36	Peppers	192.75
Blackcurrants	231.79	Collard Greens	185.21
Kiwis	101.46	Spinach	139.89
Strawberries	76.85	Red Amaranth	131.92
Kumquats	70.49	Mustard Greens	125.70
Lemons	69.45	Wuta-Tasi	122.73
Pomegranates	60.39	Rape	106.85
Oranges	53.88	Broccoli	91.90
Mangoes	50.35	Carrots	78.59
Lychees	45.34	Peas	75.89
Cherries	41.55	Chinese Cabbage	63.91
Apricots	35.09	Chinese Celery	57.85
Cantaloupes	31.86	Kidney Beans	47.81
Pineapples	29.63	Lentils	45.64
Bananas	29.44	Tomatoes	39.28
Peaches	23.42	Potatoes	38.72
Apples	20.53	White Radish	37.76
Grapes	18.48	Pumpkins	34.15
Watermelons	16.60	Eggplant	33.77
Pears	15.86	Cucumbers	30.27
		Yam	23.18
		Lettuce	17.07
		Taro	7.56
		Celery	7.23
Average	64.74	Average	72.32

## Data Availability

Data described in the manuscript was collected from published books and organized by Xuemei Zhao. All original data are showed in this manuscript and Appendix A and can be available and use freely.

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
