# Peer review of "A New Technique for Determining Micronutrient Nutritional Quality in Fruits and Vegetables Based on the Entropy Weight Method and Fuzzy Recognition Method"

_foods, 2022, doi:10.3390/foods11233844_

Round 1
Reviewer 1 Report
COMMENTS TO THE AUTHORS:
This paper has a potential to be accepted, but some important points have to be clarified or fixed before a positive action can be taken.
The following points need to be done by the authors:
1. Page 6: M&M section: “According to the classification of fruits ….” Please rewrite the sentence. It is not complete.
2. “Since fruits and vegetables are low in calories…” What is a related reference? This statement applies to some fruits and vegetables, but what about bananas? Is it still low in calories? Please check.
3. Page 6: M&M section: at the end of the first paragraph: Model? What does this word refer to? Please check.
4. Can the proposed model be used for other vitamins and minerals?
5. Check paragraphs extension in the whole manuscript.
6. English language of paper needs to be revised (English Editing Service) therefore I don't suggest word by word editions.

Author Response
ANSWER TO THE REVIEWER 1:
Thanks for your constructive suggestions and according to your comments we have adjusted some parts and adds some more details. Please check it.
- Question: Page 6: M&M section: “According to the classification of fruits ….” Please rewrite the sentence. It is not complete.
Answer: we have rewritten this sentence from the original as "According to the 2016 Chinese Food Composition Table classification for fruits and vegetables, six fruits and five vegetables were selected".
- Question:“Since fruits and vegetables are low in calories…” What is a related reference? This statement applies to some fruits and vegetables, but what about bananas? Is it still low in calories? Please check.
Answer: This question is a very important one, connected with the whole paper. It is indeed the case that some fruits and vegetables have more calories, like the banana you mentioned. However, we checked the data again and found that the banana you mentioned has fewer calories than staple foods like rice, at 89 kcal/100g and 347 kcal/100g respectively. Although the banana has a higher calorie value, it is still lower than rice and wheat. Therefore, we have changed this sentence to “Since some fruits and vegetables have lower calorie content than staple foods, such as bananas, which have higher calorie content than leafy vegetables but are still lower than rice at 89 kcal/100g and 347 kcal/100g, respectively, according to the 2016 Chinese Food Composition Table, they do not provide much energy compared to staple foods”.
- Question: Page 6: M&M section: at the end of the first paragraph: Model? What does this word refer to? Please check.
Answer: The ‘Model’ refers to the title of this section, which presents that how we built this index, but it is a little unclear. We have added more details, now it refers to ‘Vitamins and Minerals Index’
- Question: Can the proposed model be used for other vitamins and minerals?
Answer: According to the method/model we have developed, this method/model can also be used for other vitamins and minerals. With the exception of the indices for vitamin A and calcium, which are the most deficient in the daily Chinese diet, the Comprehensive Index and the Matching Index can also be used to evaluate other vitamins and minerals if sufficient data are available. The indices for vitamin A and calcium can be adapted to the needs of different regions or countries. For example, if vitamin B is the vitamin with the greatest deficiency in a region or country other than China, you only need to change the index for vitamin A to the index for vitamin B in the same way.
- Question: Check paragraphs extension in the whole manuscript.
Answer: we have checked the whole manuscript, please check it.
- Question: English language of paper needs to be revised (English Editing Service) therefore I don't suggest word by word editions.
Answer: we will consider this service.

Reviewer 2 Report
Manuscript provides some latest information about fruit and vegetables nutrient/energy profile. Your work is nice effort and I would like to be it published. However, did not written about some important micronutrients like boron, in these fruits and vegetables. Literature says that B is as important as Zn and Fe, and it must be supplemented as well. Could you answer this?
Author Response
ANSWER TO THE REVIEWER 2:
Thanks for your constructive suggestions and according to your comments we have adjusted some parts and adds some more details. Please check it.
Question: Your work is nice effort and I would like to be it published. However, did not written about some important micronutrients like boron, in these fruits and vegetables. Literature says that B is as important as Zn and Fe, and it must be supplemented as well. Could you answer this?
Answer: First of all, thank you for your recognition. As you said, some vitamins and minerals are just as important as the ones we are looking at, like the B you mentioned. It is indeed very important to evaluate every vitamin and mineral that is necessary for our body. However, there are some limitations to this work. First, there is no RNI and AI for B in China. Secondly, there is no value for B for different types of food in the Chinese food composition database. Therefore, due to the lack of relevant data to support this work, we decided to use the data we can find at this stage. However, we can also use this method/model to assess other important vitamins or minerals if enough data is available. In summary, this method focuses only on the method itself. It can be adapted for other countries in different situations. If the data collection is further completed, it can be analysed in other countries and regions. This is only a preliminary investigation/exploration.